# Migratory Engineering of T Cells for Cancer Therapy

**DOI:** 10.3390/vaccines10111845

**Published:** 2022-10-31

**Authors:** Stefanos Michaelides, Hannah Obeck, Daryna Kechur, Stefan Endres, Sebastian Kobold

**Affiliations:** 1Division of Clinical Pharmacology, Department of Medicine IV, University Hospital, Ludwig Maximilian University (LMU) of Munich, Lindwurmstrasse 2a, 80337 Munich, Germany; 2German Cancer Consortium (DKTK), Partner Site Munich, Pettenkoferstrasse 8a, 80336 Munich, Germany; 3Einheit für Klinische Pharmakologie (EKLiP), Helmholtz Zentrum München, German Research Center for Environmental Health (HMGU), Ingolstädter Landstrasse 1, 85764 Neuherberg, Germany

**Keywords:** adoptive cell therapy, CAR T cells, infiltration, cellular engineering, immunotherapy

## Abstract

Adoptive cell therapy (ACT) and chimeric antigen receptor (CAR) T cell therapy in particular represents an adaptive, yet versatile strategy for cancer treatment. Convincing results in the treatment of hematological malignancies have led to FDA approval for several CAR T cell therapies in defined refractory diseases. In contrast, the treatment of solid tumors with adoptively transferred T cells has not demonstrated convincing efficacy in clinical trials. One of the main reasons for ACT failure in solid tumors is poor trafficking or access of transferred T cells to the tumor site. Tumors employ a variety of mechanisms shielding themselves from immune cell infiltrates, often translating to only fractions of transferred T cells reaching the tumor site. To overcome this bottleneck, extensive efforts are being undertaken at engineering T cells to improve ACT access to solid tumors. In this review, we provide an overview of the immune cell infiltrate in human tumors and the mechanisms tumors employ toward immune exclusion. We will discuss ways in which T cells can be engineered to circumvent these barriers. We give an outlook on ongoing clinical trials targeting immune cell migration to improve ACT and its perspective in solid tumors.

## 1. Introduction

A paradigm change in cancer has shifted the focus of therapeutic targeting from cancer to immune cells. Groundbreaking work in basic tumor immunology has set the foundation for the immune system as the natural effector of cancer prevention and control [1]. Along these lines, cancer immunotherapy unleashing certain types of immune cells (T cells) has become the standard of care in a growing number of clinical situations and diseases [2,3,4,5]. The effective recognition of cancer cells by the immune system can be lost or suppressed during cancer progression, calling for strategies either reverting suppression or restoring recognition [6]. Most immunotherapies aim at restoring the ability of the immune system, or more precisely immune cells, to sense and eliminate cancer [7]. T cells take a key role in these developments. Checkpoint inhibitors are antibodies antagonizing suppressive molecules that can activate T cells endogenously in patients, to fight their own cancer [8]. Similarly, autologous T cells can be harvested from the patient’s peripheral blood to be genetically endowed with cancer specificity for later reinfusion with the therapeutic intention [9]. This process is broadly referred to as adoptive T cell therapy (ACT). Thus far, the only approved ACT strategy is chimeric antigen receptor (CAR) modified T cells. CAR are synthetic receptors, constituted of the antigen-binding domain of an antibody, fused to T cell activation and costimulatory domains [10]. These CAR are introduced into the T cell via viral gene transfer to stably confer the cell with target specificity. CAR targeting the B-cell-associated antigen CD19 have been tested in refractory or relapsed acute lymphatic leukemia and diverse types of B cell lymphoma [11]. Based on unprecedented efficacy leading to complete remission in a substantial number of patients treated [12,13], CAR T cells targeting CD19 have been approved by the U.S. Food and Drug Administration (FDA) and by the European Medicines Agency (EMA). A remarkable and extremely relevant aspect is that even years after infusion many patients remain disease free, indicating that these patients might in fact have been cured from their disease [14]. More recently, CAR against the plasma cell-associated antigen BCMA were developed to treat multiple myeloma patients. Again, based on substantial efficacy in patients’ refractory to other cancer therapies, CAR T cells targeting BCMA have been approved for the treatment of multiple myeloma [15].

Since T cell activation, or rather disinhibition induced by checkpoint blockade, showed great success in various solid cancer entities, it was expected that CAR T cells would be equally effective in the context of solid tumors. In sharp contrast, however, despite supportive preclinical evidence, CAR T cells have not delivered convincing clinical results in patients suffering from solid tumors so far [16]. In hematology, relapse or resistance to CAR T cell therapy is mostly due to antigen loss or the lack of T cell persistence in certain patients [17]. Although these mechanisms could certainly play a role in the failure of CAR T cells in solid oncology, the reasons appear to be radically different. In fact, we and others have identified the lack of access to tumor tissue, cancer heterogeneity, and immune suppression as hallmarks of resistance and failure in solid tumors [18]. As entering the tumor tissue comes first in the cascade of CAR T cell action, lack of access to the tumor site constitutes one of the most frequent reasons for CAR T cell failure. However, no consensus exists as to how this could be systematically overcome.

We will focus on this question by giving an overview of how T cells traffic into tumors and how this can be controlled by means of engineering. We will discuss the composition of the cellular microenvironment of tumors, point out challenges for T cell infiltration into the latter and explain ways to therapeutically tackle those challenges and change T cell infiltration and trafficking to improve therapy options. Lastly, we will also provide information on the current status of clinical testing and the development of such strategies enhancing T cell infiltration.

## 2. Cellular Infiltration to Tumor Sites and Its Challenges

### 2.1. The Basis of Cellular Migration

Migration of T cells into target tissues is essential for their activity, as T cells require target antigens and cell encounter for their function [19]. Before entering tissues, T cells circulate in the bloodstream, scanning for signals initiating extravasation. Classically, extravasation is envisioned as a five-step process: Selectin-ligand interactions tether T cells to the endothelium (1), initiating rolling along the vessel wall (2). This is followed by chemokines out of target tissues activating chemokine receptors on the rolling T cells (3). Activated T cells express integrins, which will bind to their ligands on the endothelium and lead to cellular arrest (4). Lastly, chemokine–chemokine receptor interactions lead to transendothelial migration and extravasation (5) [20,21]. This process usually takes place in high endothelial venules (HEV). Once inside target tissues, T cells follow chemokine gradients to identify their target cells and act upon them, with the release of certain chemokines determining which cells can be attracted or not based on their chemokine receptor expression pattern [20,22]. Thus, vasculature composition and selectin- as well as integrin-binding lay the foundation for any movement of immune cells. Distinct migration patterns are then determined by chemokine–chemokine receptor interactions. With over 50 human chemokines and 19 fitting receptors identified so far, cells can be precisely guided to where their action is required. This kind of mechanism is physiologically utilized for example to clear wounds or infections and is conserved in the context of cancer [23,24,25].

### 2.2. T Cell Migration into Tumors

Consequently, most research on cell recruitment by tumors focuses on chemokines, with specific chemokine signatures linked to infiltration of specific T cell subsets in tumors [26]. Both cancer and surrounding cells of the tumor microenvironment (TME) secret chemokines, which steer the migration of T cells towards and within the tumor [27,28,29]. Generally, T cell infiltration is thought to improve patient prognosis, an observation broadly confirmed in the literature [30,31,32,33,34,35,36,37,38,39]. However, the existence of pro-tumoral, immunosuppressive T cell subsets was shown as early as 2004 [40], and a more in-depth analysis of the tumor immunome reveals the large impact different T cell compositions can have on tumor progression and anti-tumor immunity [41].

The process of access to a given tumor site varies from T cell subset to T cell subset with different functional outcomes. Figure 1 provides a summary of the chemokine–chemokine receptor interactions required for specific T cell subsets to enter tumors and their functional outcome. A more in-depth analysis has already been given elsewhere [41,42,43,44].

However, as already outlined, chemokines are not the only regulators of migration. Tumor vasculature both in its density and also in its molecular signature has been shown to impact T cell infiltration in tumors [57,58,59]. Tumors with a higher density of HEVs will attract more cells than tumors with aberrant vasculature formation, while endothelium can express different ligands either stimulating or inhibiting extravasation of immune cells [57,58,59,60]. The signatures of integrins, selectins, and their ligands can also alter immune infiltration in tumors, as they are essential for stabilizing leukocyte rolling up to extravasation [61,62]. Lastly, tumor ECM density can provide a physical barrier to immune cell movement and stand in its way even after extravasation is complete [63].

Through varying the above-mentioned parameters, tumors can steer cellular infiltration to create environments beneficial to their growth. Thereby, immune-inflamed, immune-excluded, and immune-deserted tumors are distinguished [64]. In cancer immunotherapy, tackling the process of immune exclusion remains a major challenge. In order to tackle it efficiently, the variety of mechanisms by which a tumor can exclude certain immune cells while enriching others needs to be understood. We will give an overview of how tumors can act to exclude immune cells from their microenvironment before discussing ways of circumventing those exclusion mechanisms.

### 2.3. Immune Exclusion Mechanisms by Tumors

Any T cell aimed to infiltrate a tumor site reaches there through the bloodstream, initiates extravasation via selectin, chemokine, and integrin binding, follows chemokine gradients into the tumor tissue, and ultimately needs to overcome any further physical obstacles to find a cancer cell for engagement. Tumors tackle each of these axes to either fully prevent immune infiltration or enrich for specific, pro-tumoral cell subsets in their environment. In the following, we will discuss the contribution of these aspects in preventing the successful infiltration of anti-tumoral T cells.

#### 2.3.1. Tumor Vasculature

Unevenly formed, often leaky, and collapsed blood vessels, are a hallmark of many tumors. These can promote hypoxia within tumors and lead to poor trafficking of drugs and immune cells to the tumor core [65]. Hypoxic areas hinder T cell migration and movement towards tumor cells, creating immune deserted niches [66]. Vessel normalization, aiming at re-structuring tumor vessels and thus allowing better perfusion of tumors, has been proposed as a mechanism to increase immune cell infiltration into tumors. It appears to be paralleled by Th1 cell function and increases infiltration of both CD4+ and CD8+ T cells within tumors [67].

It is not only through leakiness that tumor vessels prevent T cells from entering their stromata. Endothelial signaling can have large impacts on T cell extravasation as well. Endothelin B receptor (ET(B)R) signaling in the endothelium, for example, leads to NO synthesis, which prevents T cell adhesion to the endothelium and subsequent transendothelial migration and extravasation. ET(B)R is overexpressed in the endothelium of human ovarian cancer, inhibiting T cell infiltration into the tumor, an effect reversible through treatment with the ET(B)R neutralizing agent BQ-788 [68]. Additionally, overexpression of Fas ligand (FasL) in the vasculature of different human and murine tumors is associated with weaker CD8+ T cell infiltration. Upon binding FasL, CD8+ T cells undergo apoptosis, an effect not seen on regulatory T cells, perhaps because of their expression of apoptosis inhibitors. Consequently, Tregs are recruited preferentially to tumors with high FasL in their vasculature [69].

#### 2.3.2. Integrins and Selectins

The role of integrins and selectins in tumor progression has mainly been studied regarding their expression on tumor cells and its impact on tumor metastasis, survival, and proliferation through inter- and intracellular signaling [61]. Little focus has so far been put on how differential integrin and selectin ligand expression in the tumor vasculature contributes to immune exclusion mechanisms. Nevertheless, integrin composition has been shown to alter the immunome of human tumors, thereby impacting patient prognosis [62,70]. Tumors grown in selectin ligand deficient mice are more resistant to ACT than tumors grown in WT mice due to poor infiltration by transferred T cells [71]. Additionally, altered LFA-1 signatures on T cells have been shown to hamper T cell motility in chronic lymphatic leukemia, a mechanism perhaps also used in other malignancies [72]. Lastly, epithelial loss of p53 and the αv integrin genes leads to the development of squamous cell carcinoma in mice, an effect partially based on reduced immune cell infiltration perhaps due to missing integrin signatures [73]. Altogether, selectin and integrin signatures seem to impact immune cell infiltration, but their effect remains poorly studied. More research is necessary to understand how they might promote immune exclusion in tumors.

#### 2.3.3. Chemokine–Chemokine Receptor Axes

As already discussed, chemokines are among the main mediators of T cell recruitment in human cancer. A very simple mechanism of immune cell exclusion is a chemokine–chemokine receptor mismatch, with chemokines produced by tumors and their TME not recruiting anti-tumor T cells. How this mismatch is enabled in the context of cancer is unclear. Recent discoveries reveal several mechanisms, such as epigenetic silencing of CCL5, CXCL9, and CXCL10 in different tumor entities [52,74] or post-translational cleavage of those same chemokines by enzymes such as dipeptidylpeptidase 4 or matrix metalloprotease-9 [75,76,77]. Cleavage of chemokines can both render them inactive, or in the case of CXCL10 even produce an antagonistic form of the chemokine. This has specifically been identified in malignant but not benign tumors, underlining the importance of such mechanisms in disease progression [77]. Other post-translational modifications can also be used to render chemokines inactive. Nitration of CCL2 (N-CCL2) through reactive nitrogen species within the tumor was shown to decrease its affinity to the receptor CCR2. CD8+ T cells express low overall CCR2 and are not sensitive enough to N-CCL2 to still infiltrate tumors; myeloid-derived suppressor cells, however, expressing high levels of CCR2, can still recognize the nitrated form of CCL2 and are thus not hindered in tumor entry, further contributing to the suppressive environment hostile to T cell function [53].

Tumors utilize the chemokine–chemokine receptor axis to form an immune-excluded TME in a variety of other ways as well. Spranger and colleagues identified a pathway in human melanoma, in which β-catenin produced by tumor cells suppresses the recruitment of CD103+ dendritic cells (DCs) by downregulating CCL3, CCL4, CXCL1, and CXCL2. CD103+ DCs are usually necessary to recruit CD8+ T cells through CXCL9; failure to promote their recruitment in patients of β-catenin high tumors renders these resistant to anti-PD-1 blockade therapy or abrogate treatment effects of ACT in murine tumor models [78,79]. Murine pancreatic adenocarcinomas have been shown to recruit myeloid-derived suppressor cells, which subsequently hinder the infiltration of effector T cells [80]. Similarly, tumor hypoxia was shown to trigger CCL28 production in ovarian cancer, recruiting CCR10+ regulatory T cells. CCR10+ regulatory T cells can counteract any inflammation initiated by hypoxia and thus prevent CD8+ T cell infiltration in hypoxic tumors [45]. Lastly, colon cancer seems to employ mechanisms targeted at recruiting tumor-specific, activated CD8+ T cells into liver metastases. There, they get primed to undergo apoptosis by interaction with formerly recruited FasL+ macrophages [81]. Functionally this mechanism promotes immune exclusion and immune suppression.

TGF-β mediated immune exclusion of T cells from human tumors is also partially mediated through the chemokine–chemokine receptor axis [82,83,84]. Hereby, TGF-β can downregulate CXCR3 expression on CD8+ T cells, thus excluding them from tumors [85]. TGF-β signaling can also initiate the formation of cancer-associated fibroblasts. Those in turn promote high CTLA-4 expression on T cells, leading to cell clustering and limiting the movement of T cells into the tumor [86,87]. This mechanism does not precisely target chemokines, yet it still hampers cellular motility. While the exact mechanism of immune exclusion might very well be a combination of both, these studies shed an interesting light on the number of different ways one single agent can lead to immune exclusion.

#### 2.3.4. Extracellular Matrix

Through an unusually stiff and dense ECM, tumors can prevent cellular movement towards and within them. A study in 2012 showed, that T cells within a tumor preferentially migrate to the tumor stroma rather than tumor islets, identifying a higher density of the ECM in tumor islets as the potential mechanism behind this [63]. High ECM density also inhibits migration into tumors, as stiffening and ECM density inversely correlate with T cell infiltration [88]. Central fibrosis in tumors excludes immune cells from colorectal cancer metastases [33,89]. Recently it was shown that even collagen fiber alignment can aid in excluding CD3+ T cells from tumors [90]. Additionally, besides solely inhibiting infiltration, a high ECM density has even been shown to hamper the intra-tumoral proliferation of cytotoxic T cells [91].

Altogether, many different mechanisms are used by tumors to exclude immune cells or preferentially enrich for certain immune cell subsets using axes decisive for cellular migration, in a system whose complexity we only begin to understand.

## 3. Therapeutically Altering T Cell Infiltration in ACT

One of the biggest challenges for ACT is to circumvent those hurdles and infiltrate tumors in large enough numbers for a significant anti-tumor effect. Therefore, several ways to alter a tumor´s cellular composition to a more anti-tumoral microenvironment, exist. Here, we want to provide an overview of the pre-clinical research targeted at altering T cell infiltration into tumor sites, mainly focusing on ways to increase the infiltration of adoptively transferred T cells. Generally, three different principles can be distinguished:(A)Migratory engineering of T cells, which consists of direct genetic engineering of T cells to improve their migratory capacity towards the tumor;(B)Altering the injection site of ACT to tailor tumor infiltration mechanically to the desired site;(C)Indirect engineering methods aid T cell trafficking by mechanistically altering the tumor microenvironment and ECM.

In this review, we will mainly be focusing on the direct migratory engineering of T cells and the impact of different ACT injection sites. For reading on indirect migratory engineering methods through alteration of the TME and ECM we recommend recent extensive work by us and others [24,43].

### 3.1. Direct Migratory Engineering of T Cells to Alter Tumor Trafficking

#### 3.1.1. Ectopic Chemokine Receptor Expression

As previously discussed, chemokines are among the main mediators of T cell trafficking to tumor sites. A strategy developed to overcome the problem of mismatch between chemokine receptors and tumor-chemokine in ACT is the ectopic introduction of tumor-chemokine-tailored chemokine receptors on T cells by transduction. This was first performed by Kershaw and colleagues using the chemokine receptor CXCR2 to improve migration towards CXCL1-secreting human melanoma cell lines in 2002 [92]. While only showing the feasibility of this approach in vitro, their study laid the foundation for many more investigating the use of chemokine receptors to improve tumor infiltration in ACT. To date, ectopic chemokine receptor expression for improved CAR T cell trafficking into tumors has been shown to be a viable strategy for the chemokine receptors CCR2, CCR4, CCR8, CXCR1, CXCR2, CXCR5, CXCR6 and CX3CR1 in different tumor entities [16,93,94,95,96,97,98,99,100,101,102,103,104,105]. Interestingly, some studies also report a stronger cytolytic effect and interferon-γ release of chemokine receptor transduced T cells, suggesting stronger T cell activation through chemokine receptor signaling [98,101]. Additional to interferon-γ release, chemokine receptor expression of select receptors can steer CAR T cells to interact with DCs [103], an effect shown to be essential for T cell activity and proliferation within solid tumors [28]. Chemokine expression however differs massively between different malignancies and even patients. This poses a yet unsolved problem to the application of chemokines receptors in ACT as those need to be tailored to a tumor‘s chemokines prior to treatment.

A possible avenue to circumvent this limitation could be to synthesize chemokine receptors to react to stimulants other than chemokines. This was shown to be possible both in the generation of a photoactivable chemokine receptor and of a chemokine receptor reacting to the chemical “clozapine-N-oxide” (CNO) [106,107]. That way, cellular migration can be steered externally, to a light source or to CNO-releasing beads as shown in these two studies, or in principle also to any other target. Especially the use of a photoactivable receptor could be beneficial in tumor treatment, as it was shown to be successfully applied in murine melanoma models [107]. While so far only experimental and theoretically only of use in dermatological malignancies, it provides an elegant method of steering T cells into tumors.

#### 3.1.2. Degradation of the Extracellular Matrix

A more broadly adaptable method of steering T cells into tumors, not relying on tumor-specific chemokines or similar, is the approach of equipping T cells with enzymes to degrade the ECM of tumor tissue. This encompasses the approach of ectopically expressing heparanases or hyaluronidases—enzymes that degrade polypeptides in the ECM—on CAR T cells, allowing them to navigate easier through dense tumor stroma [108,109,110]. Equipping GD2 CAR T cells with heparanase led to improved CAR T cell infiltration and anti-tumor effect in several different in vivo xenograft models. Crucially, no pathological CAR T cell accumulation in other tissues was observed [108]. Expression of the hyaluronidase PH20 follows the same principle and could enhance tumor infiltration by CAR T cells in murine models of gastric and liver cancer [109,110]. Thus, while not as extensively studied as the usage of chemokine receptors, early findings point towards a high potential of this approach combined with a broader adaptability to different entities and thus a potentially easier application in the clinic.

#### 3.1.3. Altering Chemokine Expression

Lastly, engineering T cells to improve infiltration into tumors does not necessarily have to be focused on helping them enter the tumor but can also focus on recruiting more subsequent T cells into the tumor. This can be achieved by engineering T cells to express and secret specific chemokines, which in turn will then attract more T cells. CAR T cells engineered to constitutively secret IL-7 and CCL19, recreating the T cell zone in lymphoid organs within the tumor, improved anti-tumor immunity in murine solid tumor models. Interestingly, secretion of CCL19 recruited not only T cells, but via their receptor CCR7 also DCs, aiding T cell priming and activation within the tumor as well as infiltration [111]. More trials have since looked at this mechanism, with differing levels of success. Expression of CXCL10 upon encounter of tumor antigens in a synthetic notch receptor (synNotch) steered manner improved subsequent CXCR3-based tumor infiltration by CAR T cells [112]. The synNotch system allows gene expression of an introduced gene after target antigen encounter [113]. Here, this translates to chemokine secretion once CAR T cells get activated inside the tumor. Similarly, we previously identified a CCL1-CCR8 feedback loop through CCL1 production of CAR T cells upon their activation as essential to aid the infiltration of CCR8 overexpressing CAR T cells in tumors [16]. However, in another study, CAR T cells constitutively secreting CXCL11 do not increase infiltration of CXCL11-producing, CXCR3+ CAR T cells into the tumor, although they increase the concentration of CXCL11 within tumor stromata [114]. The reason, therefore, is most likely desensitization and internalization of the CXCR3 receptor of CXCL11-producing CAR T cells due to their constant exposure to the constitutively secreted chemokine. This suggests, that engineering of T cells to produce chemokines themselves is only feasible if

(A)The production of chemokines is not constitutive, but initiated within the tumor based on prior T cell activation;(B)The beneficial effect of secreted chemokines is also based on the recruitment of other T cell or immune cell subtypes not yet chronically exposed to their chemokine.

### 3.2. Altering the Injection Site of ACT

Moving away from genetic T cell engineering, the need for adoptive cellular therapies to migrate to and enter the tumor site can be circumvented by altering the injection site of ACT. Most clinical, as well as preclinical studies, apply cellular therapies intravenously, assuming a distribution throughout the patient’s bloodstream to be sufficient to allow entry into the tumor via transendothelial migration. As already discussed, this comes with a number of shortcomings. Several studies have shown the intratumoral or locoregional application of ACT as a way of overcoming those. Direct intratumoral injection of anti-mesothelin CAR T cells in murine pleuramesothelioma models requires 30-fold fewer CAR T cells to achieve a comparable therapeutic response to an intravenous application. Interestingly, this injection method leads to better control of distant metastases as well, potentially due to a stronger CD4+ T cell activation within the tumor and subsequent recirculation of activated T cells [115]. More studies conducted in models of malignant mesothelioma, peritoneal carcinomatosis, and even CNS tumors suggest local CAR delivery be superior to the intravenous application [116,117,118,119]. A study comparing the intratumoral, locoregional, or intravenous application of CAR T cells to treat CNS tumors found the locoregional application to be outperforming both intratumoral and systemic applications. This indicates that the application of CAR therapy is close to the tumor, while still allowing a certain extent of circulation by applying it in a spatial niche perhaps being the most beneficial [118].

Another critical enhancement of CAR T cells could be alternative delivery methods. Applying CAR T cells in biopolymer scaffolds or nitinol-coated films around resected tumors showed both better expansion and activation within relapsed tumors than just injecting them in PBS [120,121]. Herein, the benefit lies in the possibility of coating these scaffolds with activating beads or cytokines, thus ensuring the better proliferation of T cells within the tumor.

In conclusion, substantial preclinical research has been undertaken at altering cellular infiltration patterns in tumor therapy. Figure 2 provides an overview of the engineering methods employed to alter CAR T cell trafficking and on which axes of immune exclusion they target. Notably, most approaches at migratory engineering target immune exclusion via the chemokine–chemokine receptor axis, calling for a higher focus to be put on the other axes of immune exclusion as well.

Preclinical success could be achieved for many of the detailed approaches, particularly when it comes to specifically aid CAR T cells to reach tumor tissue. The importance of increased migration and tumor infiltration becomes apparent when looking at outcomes of these preclinical trials, with increased infiltration of CAR T cells nearly always associated with slowed tumor growth and increased survival in in vivo models. However, the transfer of these approaches to the clinic is essential. We now want to provide an overview of the current clinical status of CAR T cell therapies and give insight into a number of studies focusing on engineering migration of CAR T cells for improved therapeutic outcomes.

## 4. Migratory Engineering in Clinical Application

### 4.1. Current Clinical Status of CAR T Cells

Currently, there are six different CAR T cell therapies approved by the U.S. Food and Drug Administration (Table 1). It should be emphasized that all approved drugs thus far have been designed for the therapy of hematological malignancies. In 2017, *tisagenlecleucel* targeting CD19 was the very first CAR T cell therapy to be approved, soon to be followed by *axicabtagene ciloleucel* also targeting CD19 later in 2017. In 2020 *brexucabtagene autoleucel* was approved, and in 2021, *Lisocabtagene maraleucel* followed with both therapies targeting again CD19. The latest CAR T cell therapies to be approved were *Idecaptagene vivleucel* and *ciltacabtagene autoleucel* in 2021 and early 2022, both targeting B cell maturation antigen (BCMA) to treat multiple myeloma. The underlying studies for the approval of these therapies were promising and showed substantial results in some patients. However, previously described problems of CAR T cells are also reflected by the actual clinical benefit in patients. *Tiagenlecleucel* for example gave an overall response rate (ORR) of 82% showing that 18% of treated patients did not respond to treatment at all with similar or worse data for all other approved CAR T cell therapies. Only the most recently approved *ciltacabtagene autoleucel* achieved an ORR of 98%. Additionally, even then, it is important to remember that ORR is only a transient reflection of benefit and most patients will progress or relapse later on. Thus, the clinical performance of CAR T cells can still be improved, even in the so far successfully treated hematological malignancies.

There are currently 1016 studies on CAR T cell therapy active worldwide (clinicaltrials.gov, data cut-off on 25 August 2022). 975 of these studies are focusing on cancer therapy. Of those, 56 studies have been completed while the others are in different stages of trial progress. 73% of the registered trials are focused solely on hematological malignancies but there is also a growing number of trials focusing on solid tumors (27%). Shown in Figure 3 is an even distribution of investigated conditions in solid tumors focusing on malignancies of gastrointestinal, pancreatic, lung, breast, and female reproductive tissues. The larger fraction of studies targeting hematological malignancies reflects the difficulties in achieving clinical success using CAR T cell therapy in solid tumors.

Through further research, we were able to filter out those studies that particularly target T cell engineering to enhance the infiltration and migration of CAR T cells. To give an overview of which of the previously described options for improving CAR T cell trafficking are therapeutically used, we have selected a few exemplary studies.

### 4.2. Direct Migratory Engineering of T Cells to Alter Tumor Trafficking

Studies targeting direct migratory engineering mostly employ ectopic expression of chemokine receptors on CAR T cells. CXCR5 and CXCR2 seem to be popular targets for this approach. CXCR5-expressing anti-EGFR CAR T cells for the treatment of non-small cell lung cancer are trialed in two different phase 1 clinical trials (NCT04153799, NCT05060796). These studies are based on recent preclinical work, showing CXCR5 to guide migration into CXCL13-high tumor tissue [99]. Additionally, CXCR2, the physiological receptor for CXCL8 has been shown to improve CAR T cell therapy preclinically [97], now being tested in a phase 1 study targeting glioblastoma (NCT05353530) where an anti-CD70 CAR T cell is additionally modified to express CXCR2. Another larger phase 2 trial combines CXCR2 and nerve growth factor receptor (NGFR) in CAR T cells to treat melanoma (NCT01740557). Lastly, CCR4 has also been shown to be relevant for T cell migration in tumors [94]. This is exploited in a phase 1 clinical trial, where CAR T cells directed against CD30 are additionally modified to express CCR4 to help the cells move to cancerous regions in the patient’s body (NCT03602157). However, this study only focuses on lymphomas and lymphatic diseases, thus not quite targeting solid tumors. Interestingly, the number of chemokine receptors trialed remains quite low, not reflecting the extensive preclinical research in this area. It remains to be seen, whether the preclinically observed benefit of chemokine receptor expression on ACT will translate to the clinic.

In two different phase 1 studies, another attempt at direct migratory engineering has been used to aid CAR T cells to infiltrate tumors. Herein, anti-CD19 CAR T cells engineered to secret IL-7 and CCL19 are used to treat diffuse large B cell Lymphomas (NCT04381741, NCT04833504), aiming at promoting tumor infiltration, accumulation, and survival of CAR T cells in cancerous tissue [111].

The last described approach of direct migratory engineering, namely degradation of the tumor ECM, could not be identified in currently registered clinical studies.

### 4.3. Altering the Injection Site of ACT

As previously described, altering the injection site of either CAR T cells or other therapeutic compounds can potentially massively improve the treatment of cancer. Some of the here-discussed trials however are focusing on reducing treatment toxicity through intratumoral CAR delivery, rather than increasing CAR T cell numbers in tumors. One example of this is a current study on glioblastoma (NCT03283631), where CAR T cells are injected directly intracerebrally into the tumor. Altering the injection site may make systemic reactions less probable. In another phase 1 clinical trial (NCT01818323), investigators are intratumorally injecting immunotherapy to treat squamous cell carcinoma of the head and neck, after this approach has proven to reduce cytotoxic side effects in tumor-bearing mice [127].

However, a small portion of trials also focuses on treating solid tumors locally to enhance T cell accumulation at the tumor site. An example of this would be the following phase 1 and 2 clinical trials in which researchers injected anti-GPC3 CAR T cells directly into the tumor of patients suffering from hepatocellular carcinoma (NCT03130712, NCT04951141).

Interestingly, the intratumoral application is not the only other route apart from intravenous injection. Intraperitoneal injections can also improve immunotherapy [119]. This was the case in the following two phase 1 studies. Here, in addition to intravenous administration, CAR T cells are also injected intraperitoneally to treat chemotherapy-refractory ovarian carcinomas (NCT05518253, NCT05420545). Additionally, in gastric cancer, especially with peritoneal metastases, the intraperitoneal application of CAR T cells seems to be beneficial (NCT03563326).

Generally, clinical trials targeting improved infiltration of CAR T cells, be it through genetic engineering or other means, remain sparse, calling for more focus to be put into this area of CAR T cell treatment in the future. As of now, no clinical evidence of its use exists yet, with most studies being at only early stages.

## 5. Conclusions

Infiltration patterns of immune cells in solid tumors can be decisive for therapeutic outcomes, particularly in adoptive cell therapies such as CAR T cell therapy. In this context, research on migratory engineering already does and will further play a major role in making the treatment of solid tumors with CAR T cells a reality. Preclinical evidence already shows its importance and potential in optimizing CAR T cell therapy, with results hopefully soon to be recapitulated in the clinic.

However, the complexity of cellular infiltration into tumors remains a challenge, as different axes can help improve the infiltration of CAR T cell therapy. No axis by itself stands out as a unique solution due to their reliance on one another in the migratory cascade. Future CAR T cells might, therefore, combine several of the discussed alterations to allow optimal trafficking into the tumor.

Even if this was guaranteed to happen, other challenges such as poor intratumoral proliferation and persistence, or mitigating the immunosuppressive microenvironment of solid tumors are not tackled by solely targeting the migration of CAR T cells. Substantial room for improvement of CAR T cell therapy in each of these axes remains, with preclinical and clinical studies giving hope for those improvements to soon arrive in the clinic.

Ultimately, a CAR T cell might integrate several optimizing approaches, importantly including enhanced infiltration capacities into tumors, to yield the best therapeutic outcome.

## Figures and Tables

**Figure 1 vaccines-10-01845-f001:**
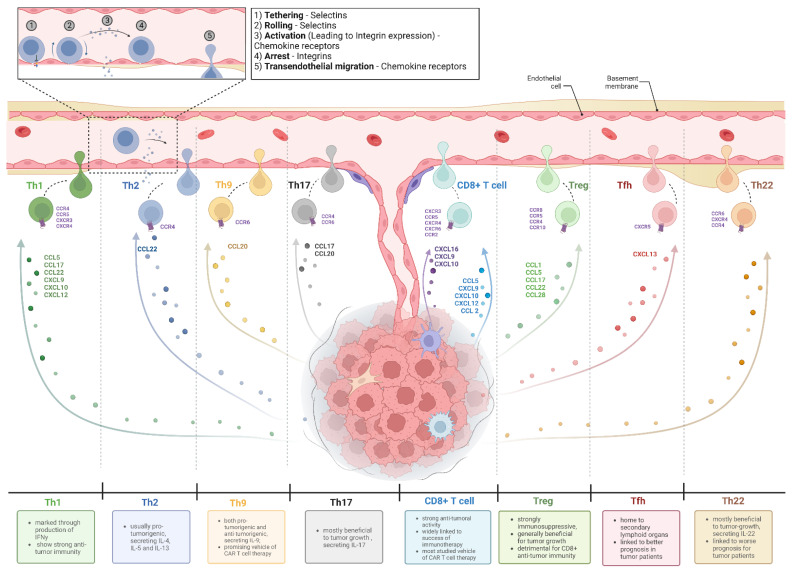
T cell migration into human tumors. Depicted are various T cell subsets which are found in solid tumors, as well as the impact each subset is regarded to have on tumor growth. To attract each T cell subset, tumors and/or cells of the TME secret a different set of chemokines which, as shown here, interacts with specific chemokine receptors, recruiting corresponding cells. The indicated chemokine–chemokine receptor interactions required for recruitment of each cell subset have been identified out of literature [26,28,45,46,47,48,49,50,51,52,53,54] and apply to different entities. Impact on tumor progression has been adapted from elsewhere [41,55,56]. Abbreviations: Th: T helper cell (correspondingly 1, 2, 9, 17, 22); Tfh: T follicular helper cell; Treg: regulatory T cell. Figure created with BioRender.com. Adapted from “Tumor Vascularization”, by BioRender.com (2022). Retrieved from https://app.biorender.com/biorender-templates (accessed on 29 September 2022).

**Figure 2 vaccines-10-01845-f002:**
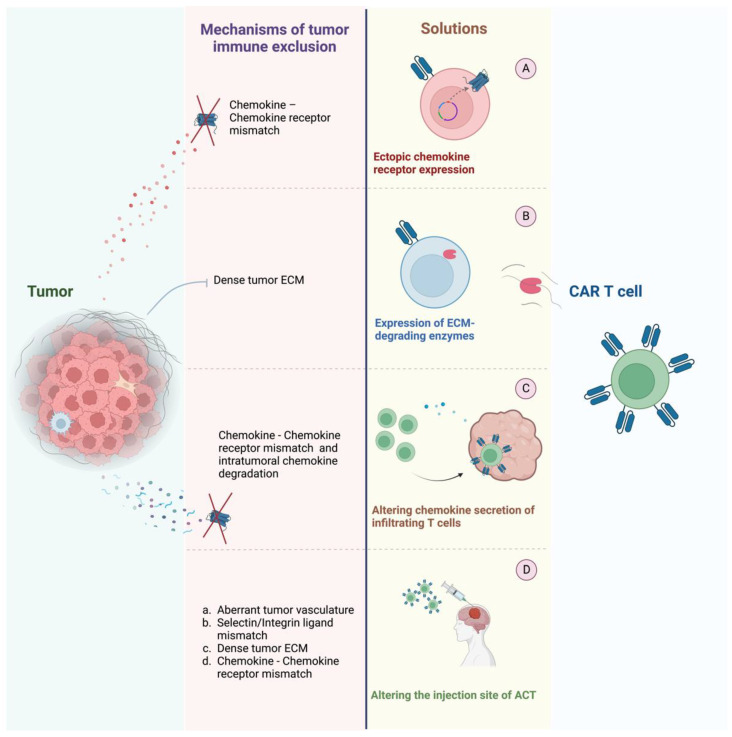
Immune exclusion mechanism by tumors and means of engineering ACT to overcome those exclusion mechanisms. Most engineering methods of ACT target the chemokine–chemokine receptor axis, with only a small proportion of trials targeting other exclusion mechanisms by tumors: (**A**) Ectopic chemokine receptor expression on CAR T cells can aid to overcome a chemokine–chemokine receptor mismatch. (**B**) ECM-degrading enzymes aid CAR T cells in infiltrating densely packed tumor beds. (**C**) Altering chemokine secretion of infiltrating T cells helps recruit more anti-tumor immune cells and subsequent CAR T cells. (**D**) Tailoring ACT delivery to the tumor site mitigates challenges posed while infiltrating into tumors. Figure created with BioRender.com.

**Figure 3 vaccines-10-01845-f003:**
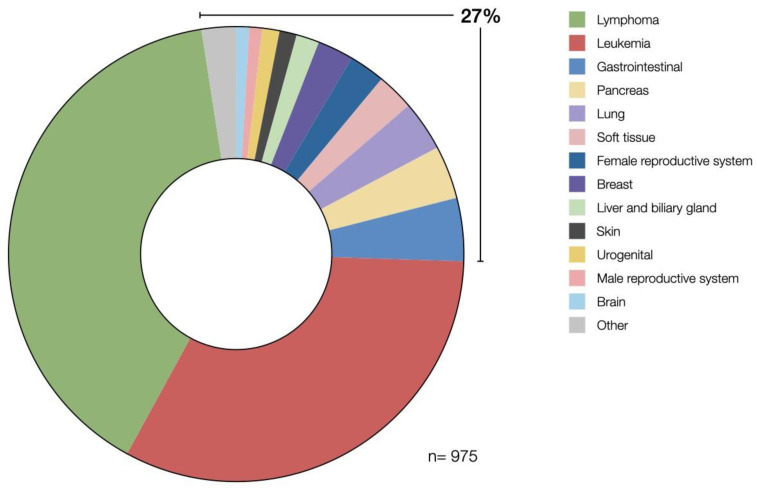
A growing number of studies registered on clinicaltrials.gov is focusing on solid tumors. The majority of studies combined 73% are still targeting various types of lymphomas and leukemia. Others include several studies on malignancies of the following tissues: adrenal gland, salivary gland, thyroid, eye, head and neck, and germ cells.

**Table 1 vaccines-10-01845-t001:** FDA-approved CAR T cell therapies thus far.

Name (Trade Name)	Target Antigen	Indication *	Underlying Trial Name and Number	Clinical Benefit	Approval Date
Tisagenlecleucel (KYMRIAH)	CD19	Acute lymphoblastic leukemia (ALL) large B-cell lymphoma including diffuse large B-cell lymphoma (DLBCL), high-grade B-cell lymphoma, and DLBCL arising from follicular lymphoma	ELIANA NCT02435849 [12] JULIET NCT02445248 [13]	82% ORR 50% ORR	30 August 2017 extension 1 May 2018
Axicabtagene ciloleucel (YESCARTA)	CD19	Large B-cell lymphoma including diffuse large B-cell lymphoma (DLBCL), primary mediastinal large B-cell lymphoma high-grade B-cell lymphoma, and DLBCL arising from follicular lymphoma	ZUMA-1 NCT02348216 [122] ZUMA-7 NCT03391466 [123]	72% ORR No ORR public yet	18 October 2017 Extension 1 April 2022
Brexucabtagene autoleucel (TECARTUS)	CD19	Mantle cell lymphoma (MCL)	ZUMA-2 NCT02601313 [124]	87% ORR	24 July 2020
Lisocabtagene maraleucel (BREYANZI)	CD19	Large B-cell lymphoma including DLBCL, primary mediastinal large B-cell lymphoma, high-grade B-cell lymphoma, and follicular lymphoma grade 3b	TRANSCEND NCT02631044 [125]	73% ORR	5 February 2021
Idecaptagene vivleucel (ABECMA)	B-cell maturation Antigen (BCMA)	Multiple myeloma	KARMMA NCT03361748 [15]	72% ORR	26 March 2021
Ciltacabtagene autoleucel (CARVYKTI)	BCMA	Multiple myeloma	CARTITUDE-1 NCT03548207 [126]	97.9% ORR	28 February 2022

* All diseases must be relapsed or refractory. All therapies consist of a single dose of intravenously applied CAR T cells on day 0 after complete lymphodepletion.

## Data Availability

Not applicable.

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
