# Peer review of "Migratory Engineering of T Cells for Cancer Therapy"

_vaccines, 2022, doi:10.3390/vaccines10111845_

Round 1

Reviewer 1 Report

Michaelides et. al. addressed a highly relevant topic in the field of immunotherapy for cancer, that is, modified T cell transfer and specifically discussed a particular problem found on this type of immunotherapy which is how to improve access to tumor by the transferred T cells.

Authors presented an exhaustive review of vast numbers of papers and current trials. Also, authors described how tumors and the tumor microenvironment prevent immune cells to infiltrate into tumor stroma which limits their effectiveness. This established a solid context for readers.

However, minor changes should be addressed:

line 87 references must be included before the period.

In figure 1, blue box (CD8 T cells) the word Most is the only one with capital M

Authors should make consistent the term dendritic cells, line 284 reads Dendritic Cells and line 323 reads dendritic cells, line 210 DCs

line 438 authors wrote others includes..it should read others include

Author Response

Reviewers comments:
Michaelides et. al. addressed a highly relevant topic in the field of  immunotherapy for cancer, that is, modified T cell transfer and specifically discussed a particular problem found on this type of immunotherapy which is how to improve access to tumor by the transferred T cells.
Authors presented an exhaustive review of vast numbers of papers and current trials. Also, authors described how tumors and the tumor microenvironment prevent immune cells to infiltrate into tumor stroma which limits their effectiveness. This established a solid context for readers.

We thank the reviewer for this comment and their appreciation of the relevance of our article.

However, minor changes should be addressed:

line 87 references must be included before the period.

In figure 1, blue box (CD8 T cells) the word Most is the only one with capital M

line 438 authors wrote others includes..it should read others include

We thank the reviewer for their attention to detail and for pointing us towards these mistakes. We agree that all the above stated should be addressed and have fixed each of them accordingly. In Figure 1, the word “Most” has been
changed to lower case (“most”) to make it consistent with the rest of the figure.

Authors should make consistent the term dendritic cells, line 284 reads Dendritic Cells and line 323 reads dendritic cells, line 210 DCs

We agree with the reviewer that this is a valid point and proceeded to use the abbreviation “DCs” for dendritic cells throughout the whole manuscript, detailed for the first time in line 210. This way, we hope to keep the reading more
concise and targeted.

Reviewer 2 Report

In the review article by Michaelides et al., the authors summarized the possible causes of CAR T cell failure in solid tumor therapy. They focused on this question by giving an overview of T cell trafficking into the solid tumor tissue and related immunological aspects, along with the possibilities of CAR T engineering. They also provide information on the current status of clinical testing and the development of strategies enhancing T-cell infiltration. 

The manuscript is well-written and well-organized, and the main focus points are well-summarized. The figures are all informative and help the understanding of the text.

The reference list is adequate.

Only some minor typos must be fixed.

I suggest accepting the manuscript after minor formal polishing. 

Author Response

Reviewers comments:

In the review article by Michaelides et al., the authors summarized the possible causes of CAR T cell failure in solid tumor therapy. They focused on this question by giving an overview of T cell trafficking into the solid tumor tissue and related immunological aspects, along with the possibilities of CAR T engineering. They also provide information on the current status of clinical testing and the development of strategies enhancing T-cell infiltration.
The manuscript is well-written and well-organized, and the main focus points are well-summarized. The figures are all informative and help the understanding of the text.

We thank the reviewer for this comment.

The reference list is adequate.

Only some minor typos must be fixed.

We agree with the reviewer, that minor typos occurred throughout the manuscript. To address this, we proof-read the manuscript with the help of a native english speaker and have identified some errors, which were fixed. We hope this issue is hereby resolved.

I suggest accepting the manuscript after minor formal polishing.

We thank the reviewer for their recommendation.